REVIEW-SYMPOSIUM

# Sodium channels and the ionic microenvironment of breast tumours

Theresa K. Leslie[1,2] 🆔 and William J. Brackenbury[1,2] 🆔

[1] *Department of Biology, University of York, Heslington, York, UK*
[2] *York Biomedical Research Institute, University of York, Heslington, York, UK*

Handling Editors: Peying Fong & Helle Praetorius

The peer review history is available in the Supporting information section of this article (https://doi.org/10.1113/JP282306#support-information-section).

**Abstract**  Cancers of epithelial origin such as breast, prostate, cervical, gastric, colon and lung cancer account for a large proportion of deaths worldwide. Better treatment of metastasis, the main cause of cancer deaths, is therefore urgently required. Several of these tumours have been shown to have an abnormally high concentration of $Na^+$ ($[Na^+]$) and emerging evidence points to this accumulation being due to elevated intracellular $[Na^+]$. This poses intriguing questions about the cellular mechanisms underlying $Na^+$ dysregulation in cancer, and its

Theresa Leslie completed a VetMB at the University of Cambridge in 2007 and practiced as a veterinary surgeon for several years before embarking on a PhD in Will Brackenbury's group in 2016, funded by the charity Breast Cancer Now. She recently graduated from her PhD. Her project focused on the role of voltage-gated sodium channels in regulating breast cancer invasion and metastasis. **Will Brackenbury** is a Senior Lecturer in Biomedical Sciences at the University of York. Following a PhD in Cell Physiology at Imperial College London, and a postdoc in Pharmacology at the University of Michigan, he started his ion channel research laboratory at York in 2011 as an MRC Fellow. His research centres on sodium channel signalling in solid tumours.

pathophysiological significance. Elevated intracellular $[Na^+]$ may be due to alterations in activity of the $Na^+/K^+$-ATPase, and/or increased influx via $Na^+$ channels and $Na^+$-linked transporters. Maintenance of the electrochemical $Na^+$ gradient across the plasma membrane is vital to power many cellular processes that are highly active in cancer cells, including glucose and glutamine import. $Na^+$ channels are also upregulated in cancer cells, which in turn promotes tumour growth and metastasis. For example, ENaC and ASICs are overexpressed in cancers, increasing invasion and proliferation. In addition, voltage-gated $Na^+$ channels are also upregulated in a range of tumour types, where they promote metastatic cell behaviours via various mechanisms, including membrane potential depolarisation and altered pH regulation. Together, recent findings relating to elevated $Na^+$ in the tumour microenvironment and how this may be regulated by several classes of $Na^+$ channels provide a link between altered $Na^+$ handling and poor clinical outcome. There are new opportunities to leverage this altered $Na^+$ microenvironment for therapeutic benefit, as exemplified by several ongoing clinical trials.

(Received 26 May 2022; accepted after revision 11 August 2022; first published online 2 October 2022)

**Corresponding author** W. J. Brackenbury: York Biomedical Research Institute, Department of Biology, University of York, Wentworth Way, Heslington, York YO10 5DD, UK. Email: william.brackenbury@york.ac.uk

**Abstract figure legend** Mechanisms of $Na^+$ channel and transporter-dependent proliferation, migration and invasion. $Na^+$ enters through VGSC, ENaC and ASIC channels and through the SGLT2 cotransporter and the NHE1 exchanger. These mechanisms may be responsible for elevating intracellular $[Na^+]$. $Na^+$ is removed from the cell via the $Na^+/K^+$-ATPase. VGSCs depolarise the cell membrane potential ($V_m$), which leads to increased migration. VGSCs also regulate transcription of genes involved in proliferation, migration and invasion. VGSCs increase the activity of NHE1, further elevating intracellular $[Na^+]$ and extracellular $[H^+]$. This acidifies the extracellular environment and aids cellular invasion through extracellular matrix. The low extracellular pH will then affect the $Na^+$ channels, increasing the inward $Na^+$ current through these in a positive feedback mechanism which increases intracellular $[Na^+]$ and extracellular $[H^+]$.

## Introduction

Cancers of epithelial origin such as breast, prostate, cervical, gastric, colon and lung cancer, account for a large proportion of deaths worldwide. Breast cancer is the most commonly diagnosed cancer in the UK, making up 15% of total cancer cases and 7% of cancer deaths (Cancer Research UK, 2022a, 2022b). Although treatments targeting hormone receptors and human epidermal growth factor receptor 2 (HER2) have improved outcomes in oestrogen receptor (ER), progesterone receptor (PR) or HER2 positive cancers, 15−20% of breast cancers cannot be targeted with such therapies because they lack these receptors and are classed as triple negative (Brouckaert et al., 2012). Metastatic breast cancer may be managed by chemotherapy but cannot currently be cured and triple negative metastatic breast cancer has the poorest median survival of 8.8 months (Lobbezoo et al., 2013). Better prevention and treatment of metastatic breast cancer is therefore urgently required.

## $[Na^+]$ is elevated in tumours

Several types of tumours have been shown to have an abnormally high concentration of $Na^+$ (Leslie et al., 2019).

This was demonstrated in the early 1980s with flame photometry and X-ray microanalysis of freeze-dried brain sections, showing elevated $Na^+$ concentration ($[Na^+]$) in glioma allograft tumours compared to contralateral brain regions (Hürter et al., 1982). The intracellular $[Na^+]$ was also discovered to be higher in cancerous cells than in non-cancerous cells, and higher in rapidly proliferating cells than slowly proliferating cells (Cameron et al., 1980). More recently elevated tumour $[Na^+]$ has been detected using $^{23}Na$-magnetic resonance imaging (MRI) in brain, breast, uterine and prostate cancer (Barrett et al., 2018; Jacobs et al., 2009; James et al., 2022 Ouwerkerk et al., 2003, 2007) (Fig. 1*A*). It is unclear whether this elevation is due to an increased $[Na^+]$ in the intracellular or extracellular compartments. Another possibility is that it is due to an increased relative volume of extracellular fluid, since $[Na^+]$ is an order of magnitude higher in extracellular fluid (140–150 mᴍ) than in intracellular fluid (10–15 mᴍ) (Hille, 2001; Madelin et al., 2014) (Fig. 1*B*).

It is possible that the relative importance of each of these factors depends on the site of a tumour. Gliomas, for example, cause brain oedema: an increase in extracellular fluid volume via increased capillary permeability, and reduced absorption of tissue fluid into parenchymal cells (Papadopoulos et al., 2004). Such an increase in the extracellular fluid volume fraction can be estimated

by measuring the apparent diffusion coefficient (ADC) of water using diffusion-weighted $^1$H-MRI (Le Bihan et al., 1986), since it is assumed that diffusion will be greater in the extracellular environment than inside cells. In glioma, successful chemotherapy treatment inducing tumour necrosis caused an increase in ADC as well as [Na$^+$], suggesting that the increase in [Na$^+$] was due to an increased extracellular volume fraction. In many tumours, however, there appears to be a reduction in extracellular volume compared to healthy tissue. When benign breast lesions were compared to breast cancer tumours, there was an inverse correlation between [Na$^+$] and ADC (Zaric et al., 2016), with higher [Na$^+$] and lower ADC in the cancerous tumours. This would suggest that the elevation in total tissue [Na$^+$] in breast cancer is due to an increase in intracellular [Na$^+$]. Interestingly, chemotherapy treatment of breast cancer reduced tissue [Na$^+$] (Jacobs et al., 2010, 2011). This reduction of [Na$^+$] was also seen with chemotherapy treatment in a mouse model of breast cancer, with no change in ADC (James et al., 2022) (Fig. 1C). Despite these treatments leading to cell death, which would be expected to increase the extracellular fluid volume fraction, in each study the total tissue [Na$^+$] decreased. Considered together, these studies in breast cancer suggest that intracellular [Na$^+$] is elevated and may be reduced by chemotherapy treatment. This poses an intriguing question about the cellular mechanisms underlying Na$^+$ dysregulation in cancer.

## Extracellular pH is reduced in tumours

A more well-known ionic dysregulation in the tumour microenvironment is the reduction of extracellular pH seen in many tumours (Reshkin et al., 2014; White et al., 2017). This is likely to be linked to the Warburg effect; where cancer cells use glycolysis for ATP production even when oxygen is not limiting (Vander Heiden et al., 2009; Warburg, 1956). Despite the excessive production of acidic metabolites, there appears to be a strong drive for cancer cells to maintain a neutral or slightly alkaline intracellular pH, perhaps to remove inhibition of further glycolysis. In fact, intracellular alkalinisation is an early event in oncogenesis caused by induction of the viral oncogene HPV16 E7 (Reshkin et al., 2000). pH regulation is tightly linked to Na$^+$ homeostasis since many of the cellular pH regulatory mechanisms depend on the inward Na$^+$ gradient (Fig. 2). Of particular note are the Na$^+$/H$^+$ exchanger, NHE1, and the Na$^+$–bicarbonate cotransporter, NBCn1, both of which are upregulated and highly active in breast cancer (Amith et al., 2015; Boedtkjer et al., 2013; Cardone et al., 2005; McIntyre et al., 2016; Toft et al., 2021). These studies indicate that inhibition of Na$^+$-dependent pH regulatory mechanisms shows promise in treating chemotherapy-resistant hypoxic tumours.

## Na$^+$/K$^+$-ATPase

The plasma membrane potential and Na$^+$ gradient is generated by the Na$^+$/K$^+$-ATPase, which consumes a large proportion of the total cellular ATP supply. This proportion can be 40% in kidney cells and more than this in the brain (Attwell & Laughlin, 2001; Whittam & Willis, 1963), but it is not yet known in cancer cells. Na$^+$/K$^+$-ATPase activity responds to changes in

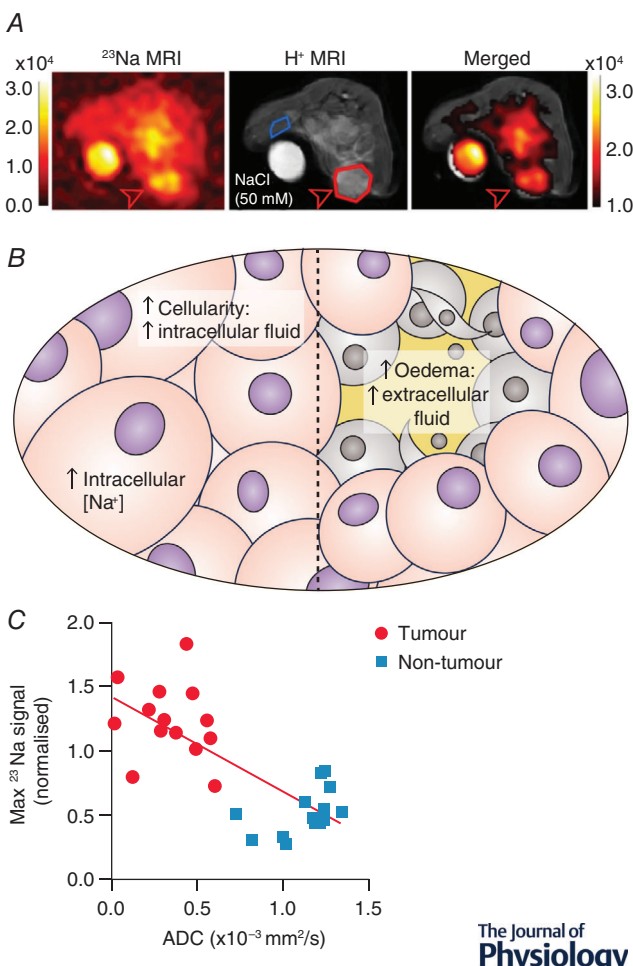

**Figure 1. Potential interplay between elevated tissue [Na$^+$] and cellularity in breast cancer**
*A*, $^{23}$Na-MRI images of a mouse mammary tumour (open red arrow), showing elevated [Na$^+$] in the tumour compared to healthy contralateral tissue (adapted from James et al., 2022). *B*, possible causes of increased tumour [Na$^+$]. Left: increased intracellular [Na$^+$] must explain the change if the cellularity is increased, and the extracellular fluid volume is decreased. Right: oedema in glioma or necrotic tumours causes an increase in extracellular fluid volume which would increase the total tissue [Na$^+$]. *C*, inverse correlation between tumour [Na$^+$] and apparent diffusion coefficient (ADC) which estimates extracellular fluid volume (adapted from James et al., 2022).

intracellular [$Na^+$] by direct binding of cytoplasmic $Na^+$ (Therien & Blostein, 2000) and via increasing activity of the salt-inducible kinase SIK1 (Sjöström et al., 2007). Reduced sensitivity of $Na^+/K^+$-ATPase to intracellular [$Na^+$] via modulation of SIK1 signalling could be responsible for the increase in intracellular [$Na^+$] seen in tumours. Indeed, SIK1 is downregulated in many cancers including breast cancer (Ponnusamy & Manoharan, 2021), supporting this possibility. Alternatively, $Na^+/K^+$-ATPase activity might be limited by ATP supply in cancer cells, leading to accumulation of intracellular [$Na^+$].

## $Na^+$-dependent nutrient import into cancer cells

Maintenance of the electrochemical $Na^+$ gradient across the plasma membrane is vital to power many cellular processes that are highly active in cancer. For example, glucose is imported via the $Na^+$–glucose co-transporter SGLT2 (Fig. 2), which is functionally active in prostate and pancreatic cancer (Scafoglio et al., 2015). This transporter aids tumour growth and metastasis, and is upregulated in lung cancer metastases compared to primary tumours (Ishikawa et al., 2001). Cancer cells have a higher glucose intake than normal cells to fuel their highly glycolytic metabolism and this is the basis of 2-deoxy-2-[$^{18}$F]fluoro-D-glucose positron emission

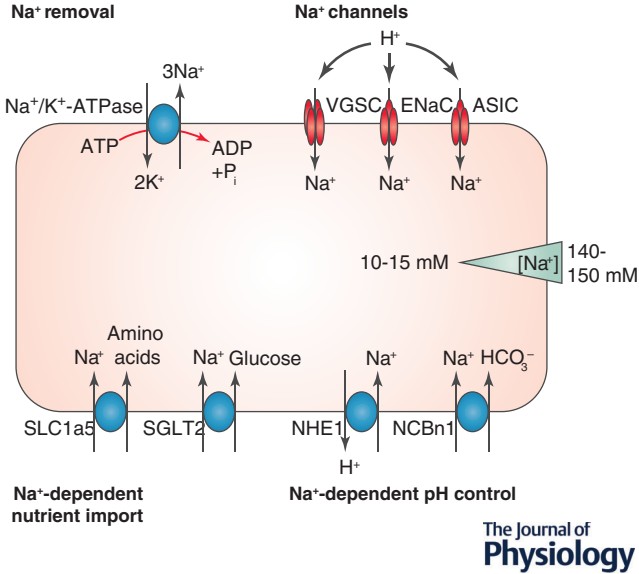

**Figure 2. $Na^+$ channels and transporters in cancer cells as discussed in this article**
$Na^+/K^+$-ATPase generates the $Na^+$ gradient across the plasma membrane. $Na^+$ channels such as VGSC, ENaC and ASIC lead to $Na^+$ entry, particularly when the extracellular pH is low. Amino acids and glucose are imported via $Na^+$-dependent transporters SLC1a5 and SGLT2. The intracellular pH is maintained at a neutral or slightly alkaline pH by $Na^+$-dependent pH regulatory mechanisms NHE1 and NCBn1 (adapted from Leslie et al. 2019).

tomography scans for detection of metastases (Czernin, 2002).

Just as cancer cells rely heavily on glucose, many are also 'addicted' to glutamine (Bhutia et al., 2015), although in healthy tissue this is a non-essential amino acid. Glutamine is important for regulation of protein synthesis via control of mammalian target of rapamycin complex 1 (mTORC1) and is vital for cell growth (Wise & Thompson, 2010). The $Na^+$ gradient is used to import glutamine and other amino acids into cancer cells via SLC1a5/ASCT2 (Fig. 2), which is upregulated and highly active in many cancer types (Dolinska et al., 2003; Hassanein et al., 2013; van Geldermalsen et al., 2016; Witte et al., 2002).

## $Na^+$ channels in cancer cells

Despite the need to maintain a $Na^+$ gradient across the plasma membrane, several $Na^+$ channels are upregulated in cancers, allowing further $Na^+$ entry into cancer cells and depleting the $Na^+$ gradient (Fig. 2). The physiological reason for this increase in $Na^+$ channels is still unclear although they promote tumour growth and metastasis via several mechanisms. One such channel is the epithelial $Na^+$ channel ENaC, which is primarily responsible for reabsorbing $Na^+$ (and therefore water) in the kidney collecting duct. ENaC protein is increased in hepatic cellular carcinoma tissue compared to matched healthy tissue and it increases proliferation, migration and invasion of hepatic cancer cells (Bondarava et al., 2009; Jin et al., 2015). ENaC is also upregulated at the mRNA level in melanoma, breast, hepatic and brain cancer cell lines (Amara et al., 2016; Bondarava et al., 2009; Kapoor et al., 2009; Yamamura et al., 2008). ENaC is closely related to the acid-sensing ion channel (ASIC) which is also overexpressed in cancers including colorectal carcinoma where it increases invasion and proliferation via the NFAT1 axis, particularly in acidic conditions such as those found in the tumour microenvironment (Zhou et al., 2017). ASICs are also involved in invasion and epithelial to mesenchymal transition, a step which is required for cancer cells to metastasize, in pancreatic cancer and hepatic cancer (Jin et al., 2015; Zhu et al., 2017). Both ENaC and ASIC increase permeability to $Na^+$ in low extracellular pH (Boscardin et al., 2016; Collier & Snyder, 2009), and could be responsible for increasing intracellular [$Na^+$], particularly in acidic tumours.

Another group of $Na^+$ channels of importance to oncogenesis are the voltage-gated $Na^+$ channels (VGSCs). These are classically responsible for depolarising the cell membrane potential, which initiates action potentials in electrically excitable cells such as neurons and myocytes. VGSCs have been identified in breast, colon, prostate, ovarian and lung cancers amongst others (Diaz et al., 2007; House et al., 2010; Laniado et al., 1997;

Roger et al., 2003, 2007). VGSC activity increases directional migration and invasion of breast, prostate and lung cancer cells through extracellular matrix (Brackenbury et al., 2007; Fraser et al., 2005; Grimes et al., 1995; Laniado et al., 1997; Roger et al., 2003, 2007; Smith et al., 1998). *In vivo*, VGSC activity increases xenograft breast tumour growth and metastasis (Nelson et al., 2015b). Expression of VGSC $\alpha$-subunit mRNA is upregulated in prostate, breast, cervical and ovarian cancer (Diss et al., 2005; Gao et al., 2010; Hernandez-Plata et al., 2012; Yang et al., 2012) and protein expression of the cardiac VGSC $\alpha$-subunit $Na_v1.5$ is associated with increased lymph node invasion in breast cancer (Fraser et al., 2005; Nelson et al., 2015b; Yamaci et al., 2017).

## Mechanisms of VGSC-induced migration and invasion

Several mechanisms have been identified to explain VGSC-induced invasion of cancer cells. These describe actions of both pore-forming $\alpha$-subunits and auxiliary $\beta$-subunits. VGSC $\alpha$-subunits are large ($\sim$270 kDa) proteins containing four homologous domains, each with six transmembrane $\alpha$-helices (Catterall, 2014). The $Na^+$ pore opens transiently in response to depolarisation of the cell membrane potential and this event is quickly followed by channel inactivation due to movement of an intracellular inactivation loop. Sometimes these channels fail to inactivate, allowing a long-lasting $Na^+$ current into the cell. This 'persistent' current is usually 1−3% the size of the transient current but can be responsible for a significant $Na^+$ influx (Alzheimer et al., 1993; Eijkelkamp et al., 2012). The $\alpha$-subunits each have specific tissue distribution and they are classified as tetrodotoxin-sensitive or -resistant (Savio-Galimberti et al., 2012).

VGSC $\beta$-subunits are small transmembrane proteins with an extracellular immunoglobulin domain, through which they act as cell adhesion molecules. $\beta1$ is the most highly expressed isoform in breast cancer, where it is upregulated compared to healthy tissue (Chioni et al., 2009; Nelson et al., 2014). $\beta1$ increases tumour growth and metastasis in a xenograft model of breast cancer (Nelson et al., 2014). This may be due to its ability to induce functional expression of a tetrodotoxin-sensitive $\alpha$-subunit (Haworth et al., 2021), or due to cell adhesion-mediated signalling via the src family fyn kinase, an action which depends on a $Na^+$ current through the $\alpha$-subunit (Brackenbury et al., 2008, 2010). In contrast, $\beta4$ acts as a tumour suppressor, preventing migration and invasion independent of $\alpha$-subunit activity (Bon et al., 2016).

$Na_v1.5$ was shown to act as a master regulator of a network of invasion genes in colon cancer (House et al., 2010, 2015). In these studies, pharmacological elevation of

the persistent $Na^+$ current upregulated the protein kinase A/extracellular signal-regulated kinase/c-JUN/ETS-like gene 1/ETS-1 transcriptional pathway, possibly by interacting with the small GTPase Rap1, which is activated by depolarisation of the cell membrane potential. A similar mechanism was identified to control migration, whereby VGSC-induced cell membrane depolarisation activates another small GTPase, Rac1 (Yang et al., 2020). This GTPase initiates branching of the actin cytoskeleton necessary for formation of lamellipodia at the leading edge of migrating cells (Pullar et al., 2006; Wu et al., 2009). The paradigm linking depolarisation and small GTPase activation was first described for K-Ras: depolarisation was shown to alter interactions between charged phospholipids at the plasma membrane, specifically phosphatidylserine in the inner leaflet, leading to nanoclustering of anchored small GTPases and their subsequent activation (Zhou et al., 2015).

A separate mechanism for VGSC-mediated invasion is promotion of $H^+$ extrusion from breast cancer cells via increasing NHE1 activity (Brisson et al., 2011, 2013; Gillet et al., 2009). Reducing the extracellular pH via this system activates enzymes which degrade the extracellular matrix, particularly the cysteine cathepsins. These studies showed that extracellular pH is particularly low in caveolae (lipid-raft-associated invaginations of the plasma membrane), which leads to effective breakdown of extracellular matrix around the invadopodia. The mechanism underlying this promotion of NHE1 activity is not obvious since $Na^+$ entry would be expected to reduce the driving force for NHE1 $H^+$ extrusion. Interestingly, $Na_v1.5$ colocalised with NHE1 in these lipid rafts, which prompted the authors to propose that $Na_v1.5$ might increase NHE1 activity via an allosteric interaction. They also showed that low pH caused an increase in NHE1-mediated $Li^+$ uptake into cells, and the pH sensitivity was stronger in the presence of $Na_v1.5$ (Brisson et al., 2013). The authors concluded that $Na_v1.5$ increases the pH sensitivity of NHE1. An additional possibility is that low pH increases $Li^+$ uptake via $Na_v1.5$, since this channel has also been shown to be pH sensitive (Onkal et al., 2019). Indeed, the $H^+$ efflux was dependent on $Na^+$ conductance through $Na_v1.5$ (Brisson et al., 2011), which raises the question of whether the $Na^+$ influx itself might be involved in increasing $H^+$ extrusion through NHE1, rather than a direct interaction between the $Na_v1.5$ and NHE1 proteins.

*SIK1* acts as a tumour suppressor and is downregulated in many cancers including breast cancer. Knock-down of SIK1 increases VGSC-induced invasiveness in breast cancer cells (Gradek et al., 2019). Given that high intracellular $[Na^+]$ normally activates SIK1, which then inhibits glycolysis (Ponnusamy & Manoharan, 2021), knock-down of SIK1 reduces its inhibition of glycolysis and may therefore increase $H^+$ production,

thus promoting low pH-dependent invasion. $Na^+$ entry through VGSCs may also upregulate glycolysis in cancer cells by promoting $Na^+/K^+$-ATPase activity to maintain the $Na^+$ gradient across the plasma membrane (Soltoff & Mandel, 1984; Therien & Blostein, 2000). $Na^+/K^+$-ATPase preferentially uses ATP derived from glycolysis in skeletal muscle, vascular smooth muscle and breast cancer cells (Dutka & Lamb, 2007; Epstein et al., 2014; James et al., 1996; Paul et al., 1979), so upregulation of $Na^+/K^+$-ATPase activity might be expected to increase the rate of glycolysis. Intriguingly, the non-voltage-gated isoform of the VGSC $\alpha$-subunit, $Na_x$, engages in lactate signalling in glial cells (Berret et al., 2013; Shimizu et al., 2007). This channel increases glucose uptake alongside $H^+$ and lactate production in response to elevated extracellular $[Na^+]$ (Nomura et al., 2019). $Na_x$ channels, which have substantial homology to voltage-gated isoforms of VGSCs, interact directly with the $\alpha$-subunit of the $Na^+/K^+$-ATPase to mediate this $Na^+$-dependent increase in glycolysis (Shimizu et al., 2007). Interestingly, the gene encoding $Na_x$, *SCN7A*, recently emerged as a key gene associated with tumour mutational burden in gastric cancer (Li et al., 2022). Thus, the importance of $Na^+$ channels in regulating progression across different cancer types may be greater than previously anticipated.

## Interplay between elevated intracellular $[Na^+]$ and extracellular $[H^+]$

As well as promoting extracellular acidification, VGSCs are in turn regulated by low extracellular pH. An acidic extracellular pH reduces the transient $Na^+$ current but greatly increases the persistent $Na^+$ current through VGSCs, particularly in the most commonly found isoform in breast and colon cancer cells, $Na_v1.5$ (Ghovanloo et al., 2018; Khan et al., 2002, 2006). VGSCs are not the only $Na^+$ channels which open in response to low extracellular pH, since ENaC and ASIC channels also increase $Na^+$ entry in acidic extracellular pH (Boscardin et al., 2016; Collier & Snyder, 2009). Together, VGSC-induced extracellular acidification and acid-induced $Na^+$ entry through $Na^+$ channels may produce a positive feedback mechanism leading to elevated intracellular $[Na^+]$ and extracellular $[H^+]$. This mechanism may help to explain why $[Na^+]$ is elevated in many tumours, and it may provide clues about new ways to target cancer therapeutically.

## Clinical implications

The utility of $^{23}Na$ MRI in monitoring response to chemotherapy is starting to become apparent, because this imaging modality can detect physiological changes in tumours before there has been a change in tumour size (Jacobs et al., 2010, 2011; James et al., 2022). Fast assessment of response to chemotherapy reduces the side effects associated with ineffective treatments and hastens the introduction of effective ones, so this imaging modality has the potential to improve treatment of cancer patients. In addition, VGSC expression correlates with poor prognosis in breast cancer and could therefore be used as a biomarker (Fraser et al., 2005).

Several retrospective studies indicate that treatment with local anaesthetics around the time of surgical tumour removal lengthens the disease-free interval (Djamgoz et al., 2019; Forget et al., 2019; Lopez-Charcas et al., 2021). Local anaesthetics act by inhibiting VGSCs, although their cancer-protective effects may be due to their ability to reduce the required doses of inhalational and opioid anaesthetics, which can be immunosuppressive. Another therapeutic use for VGSC inhibitors is prevention of seizures. Anti-epileptic drugs prescribed to breast cancer patients substantially improved survival after radiotherapy for brain metastases (Reddy et al., 2015). The anti-epileptic drug phenytoin also delayed tumour growth and metastasis in an *in vivo* model of breast cancer (Nelson et al., 2015a). A third type of VGSC inhibitors used clinically is the class I antiarrhythmic drugs. Ranolazine, a class 1b antiarrhythmic which inhibits the persistent $Na^+$ current, reduced metastasis in *in vivo* models of breast and prostate cancer (Bugan et al., 2019; Driffort et al., 2014). Clinical trials of drugs targeting $Na^+$ channels and transporters in cancer were reviewed in Leslie et al. (2019) and there are currently two prospective trials examining the systemic or intratumoral delivery of the local anaesthetic lidocaine in the perioperative period (NCT01916317, R. A. Badwe, 2013; NCT02786329, D. Ionescu, 2016).

## Conclusion

In conclusion, we have summarised the recent findings relating to elevated $Na^+$ in the tumour microenvironment and how this may be regulated by several classes of $Na^+$ channels and transporters. In particular, our understanding of how $Na^+$ channels, particularly VGSCs, contribute to cancer cell invasion and metastasis has grown based on recent studies, such that a picture is now emerging which may start to explain a key mechanistic link between altered $Na^+$ handling in solid tumours and poor clinical outcome. Given the accessibility and pharmacological tractability of these channels and transporters, there are opportunities to leverage the changes in $Na^+$ transport seen in cancer for therapeutic benefit, as exemplified by several ongoing clinical studies.

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

## Additional information

### Competing interests

None.

### Author contributions

T.K.L. and W.B. both contributed to the conception and design of the work and drafting the work and revising it critically for important intellectual content. Both authors approved the final version of the manuscript and agree to be accountable for all aspects of the work in ensuring that questions related to the accuracy or integrity of any part of the work are appropriately investigated and resolved. All persons designated as authors qualify for authorship, and all those who qualify for authorship are listed.

### Funding

This work was supported by Cancer Research UK (A25922) and Breast Cancer Now (2015NovPhD572).

### Keywords

breast cancer, metastasis, migration, proliferation, sodium, ion channels

### Supporting information

Additional supporting information can be found online in the Supporting Information section at the end of the HTML view of the article. Supporting information files available:

**Peer Review History**

