## [Peer Review History · The Journal of Physiology]

Sodium channels and the ionic microenvironment of breast tumours

Theresa K Leslie and William J Brackenbury

DOI: 10.1113/JP282306

Corresponding author(s): William Brackenbury (william.brackenbury@york.ac.uk)

The following individual(s) involved in review of this submission have agreed to reveal their identity: Christian Stock (Referee #1)

Review Timeline:

Submission Date:	26-May-2022
Editorial Decision:	23-Jun-2022
Revision Received:	21-Jul-2022
Accepted:	11-Aug-2022

Senior Editor: Peking Fong

Reviewing Editor: Helle Praetorius

Transaction Report:

Dear Dr Brackenbury,

Re: JP-SR-2022-282306 "Sodium channels and the ionic microenvironment of breast tumours" by Theresa K Leslie and William J Brackenbury

Thank you for submitting your invited Review-Symposium to The Journal of Physiology. It has been assessed by a Reviewing Editor and by 2 expert referees and I am pleased to tell you that it is considered to be acceptable for publication following satisfactory revision.

The reports are copied at the end of this email. Please address all of the points and incorporate all requested revisions, or explain in your Response to Referees why a change has not been made.

NEW POLICY: In order to improve the transparency of its peer review process The Journal of Physiology publishes online as supporting information the peer review history of all articles accepted for publication. Readers will have access to decision letters, including all Editors' comments and referee reports, for each version of the manuscript and any author responses to peer review comments. Referees can decide whether or not they wish to be named on the peer review history document.

I hope you will find the comments helpful and have no difficulty in revising your manuscript within 4 weeks.

Your revised manuscript should be submitted online using the links in Author Tasks Link Not Available. This link is to the Corresponding Author's own account, if this will cause any problems when submitting the revised version please contact us.

The image files from the previous version are retained on the system. Please ensure you replace or remove any files that have been revised. Your revised submission should include:

- A Word file of the complete text (including figure legends any Tables);
- An Abstract Figure (with legend in the Article file)
- Each figure as a separate, high quality, file;
- A full Response to Referees;
- A copy of the manuscript with the changes highlighted.
- Author profile. A short biography (no more than 100 words for one author or 150 words in total for two authors) and a portrait photograph of the two leading authors on the paper. These should be uploaded, clearly labelled, with the manuscript submission. Any standard image format for the photograph is acceptable, but the resolution should be at least 300 dpi and preferably more.

- A 'Cover Art' file for consideration as the Issue's cover image;
- Appropriate Supporting Information (Video, audio or data set https://jp.msubmit.net/cgi-bin/main.plex?form_type=display_requirements#supp).

To create your 'Response to Referees' copy all the reports, including any comments from the Reviewing Editor into a Word, or similar, file and respond to each point in colour or CAPITALS and upload this when you submit your revision.

I look forward to receiving your revised submission.

If you have any queries please reply to this email and staff will be happy to assist.

Yours sincerely,

Dr Peiyong Fong
Senior Editor
The Journal of Physiology
<https://jp.msubmit.net>
<http://jp.physoc.org>
The Physiological Society
Hodgkin Huxley House
30 Farringdon Lane
London, EC1R 3AW
UK
<http://www.physoc.org>
<http://journals.physoc.org>

REQUIRED ITEMS:

-Please upload separate high quality figure files via the submission form.

-Author profile(s) must be uploaded via the submission form. Authors should submit a short biography (no more than 100 words for one author or 150 words in total for two authors) and a portrait photograph of the two leading authors on the paper. These should be uploaded, clearly labelled, with the manuscript submission. Any standard image format for the photograph is acceptable, but the resolution should be at least 300 dpi and preferably more. A group photograph of all authors is also acceptable, providing the biography for the whole group does not exceed 150 words.

-It is the authors' responsibility to obtain any necessary permissions to reproduce previously published material
https://jp.msubmit.net/cgi-bin/main.plex?form_type=display_requirements#use

EDITOR COMMENTS

Reviewing Editor:

As seen in the evaluations below, your contribution has been very favourably esteemed by experts in the field. For resubmission please regard the queries from the reviewers with particular care to include and refer to your figures and to correct the relation of driving force for NHE1 in regard to the intracellular Na⁺ concentration.

Senior Editor:

Many thanks for this contribution. Both Referees and the Reviewing Editor appreciate the clarity in reviewing this timely and potentially impactful topic, and overall I agree with their collective assessments. This both encapsulates and complements the senior author's presentation at the on-line EMT symposium "New Roles for Ion Channels and Transporters in Health and Disease" very well.

The Authors will see, nonetheless, a few queries did arise that you likely will have no problem addressing. As advised by the Reviewing Editor, the key elements to direct attention regard incorporation of references to figures (per comments of Ref. 1) and clarification of the relationship of NHE1 driving force with the intracellular sodium concentration (last comment of Ref. 2).

REFEREE COMMENTS

Referee #1:

In their review entitled "Sodium channels and the ionic microenvironment of breast tumors", Leslie and Brackenbury give an overview of the increasing knowledge of the Na⁺ concentration and the expression of voltage sensitive Na⁺ channels in breast cancer tissue. The topic is timely and the information presented will be of great impact, for cell biologists and (electro)physiologists as well as clinicians. As mentioned by the authors, ²³Na MRI can be used to monitor the response to chemotherapy, and the expression level of voltage gated sodium channels (VGSCs) may serve as a biomarker, because VGSC expression correlates with malignancy/poor prognosis. The authors refer to work from their own, Mustafa Djamgoz's and Sébastien Roger's/Jean-Yves LeGuennec's group, i.e. those groups who have been fathering the concept of Na⁺ and VGSCs in breast cancer. They also describe VCSCs subunits as adhesion molecules and how VGSC activity can promote MMP-dependent invasiveness. In this context Leslie & Brackenbury do mention the interplay between VGSCs, NHE1 and the

resulting extracellular acidification promoting cathepsin activity, however, the molecular mechanism remains vague by which increased Na⁺ influx through VGSCs (causing a decrease in the Na⁺ gradient) would stimulate NHE1 (taking advantage of this Na⁺ gradient) while decreasing the "driving force" for it. It would be nice if the authors could provide more details in addition to what they mention on top of page 8: "...via an allosteric interaction...". In the original paper by Lucie Brisson et al. (2013, J Cell Sci) is written: "The change in cooperativity strongly suggests that Nav1.5 function in cancer cells allosterically increases NHE-1 activity in neutral ranges of pH". - We have learnt more about the molecular structure of NHE1 and its molecular interactions since then. Is there any new knowledge of possible molecular switches? Brisson et al. (2013) conclude from a decrease in pH_i-dependent Li⁺-uptake in shNav1.5 breast cancer cells that Nav1.5 promotes NHE1 activity. How do they exclude that Nav1.5 or other channels do not let pass Li⁺?

- In addition to the graphical abstract the authors have prepared two nice figures but do not refer to them in the text body.
- Page 3, graphical abstract: May I suggest to "reverse" the arrow pointing from "Intracellular [Na⁺]" to NHE1, so that it points from NHE1 to [Na⁺]_i? Actually, an increase in [Na⁺]_i (assuming constant pH) should slow down NHE1 activity.
- same as above: What would be the driving force for NHE1 if [Na⁺]_i and pH_e decreased? I bring up this point once again because the authors state on the bottom of page 6: "Despite the need to maintain a Na⁺ gradient across the plasma membrane, several Na⁺ channels are upregulated in cancer, allowing further Na⁺ entry into cancer cells and >>depleting the Na⁺ gradient<<".
- Page 5, bottom: "of particular note are the Na⁺/H⁺ exchanger, NHE1, and the Na⁺/bicarbonate cotransporter, NBCn1, both of which are upregulated and highly active in breast cancer (Cardone et al.,.....). This information and the references could be extended by a more recent publication from the Boedtker-group (Acid-base transporters and pH dynamics in human breast carcinomas predict proliferative activity, metastasis, and survival. Toft NJ, Axelsen TV, Pedersen HL, Mele M, Burton M, Balling E, Johansen T, Thomassen M, Christiansen PM, Boedtker E. *Elife*. 2021 Jul 5;10:e68447. doi: 10.7554/eLife.68447). In this paper, Toft and colleagues draw a more detailed and differentiated picture of NHE1 and NBCn1 in the different types of breast cancer.
- Could a low salt diet represent a good adjuvant therapy?
- page 8, 2nd paragraph, 1st sentence: should the comma be kept between "small" and "transmembrane"?
- Figure legend 1C, last line: "...causes an increased extracellular..." or "causes an increase IN extracellular...".

Referee #2:

Leslie and Brackenbury present a concise review on sodium (channels) in breast cancer. This is a timely topic with a clear prospect for translation, and the text is well written so that there are only a few points that deserve further attention/clarification.

pg. 6, 1st para: The authors refer to the salt-inducible kinase SIK1. Is there any knowledge about this kinase in (breast) cancer with its elevated Na⁺ concentration? Is it pH sensitive?

pg. 7, In4...: The authors cite several papers reporting upregulated expression of ENaC. Here a better distinction should be made between findings made in cell lines and those made in patient samples. Moreover, some of the papers only provide expression and no functional data.

pg. 8, 3rd para: The authors state that small GTPases are regulated by membrane potential depolarization. When looking up the papers I was surprised to see that the described depolarization is quite small, only a few millivolt. Can the authors speculate by which mechanism the small electrical signal is transduced into a biochemical one?

pg. 9, top: It is not quite intuitive that an increase of Na^+ should increase NHE1 activity. In terms of driving force, the opposite should be the case. Please discuss in more detail.

END OF COMMENTS

Confidential Review

26-May-2022

JP-SR-2022-282306 - The Journal of Physiology - Response to Referee comments

EDITOR COMMENTS

Reviewing Editor:

As seen in the evaluations below, your contribution has been very favourably esteemed by experts in the field. For resubmission please regard the queries from the reviewers with particular care to include and refer to your figures and to correct the relation of driving force for NHE1 in regard to the intracellular Na⁺ concentration.

Senior Editor:

Many thanks for this contribution. Both Referees and the Reviewing Editor appreciate the clarity in reviewing this timely and potentially impactful topic, and overall I agree with their collective assessments. This both encapsulates and complements the senior author's presentation at the on-line EMT symposium "New Roles for Ion Channels and Transporters in Health and Disease" very well.

The Authors will see, nonetheless, a few queries did arise that you likely will have no problem addressing. As advised by the Reviewing Editor, the key elements to direct attention regard incorporation of references to figures (per comments of Ref. 1) and clarification of the relationship of NHE1 driving force with the intracellular sodium concentration (last comment of Ref. 2).

Response:

We thank the Editors for their positive overall assessment of the manuscript. We have addressed the specific referee comments below, point-by-point.

REFEREE COMMENTS

Referee #1:

In their review entitled "Sodium channels and the ionic microenvironment of breast tumors", Leslie and Brackenbury give an overview of the increasing knowledge of the Na⁺ concentration and the expression of voltage sensitive Na⁺ channels in breast cancer tissue. The topic is timely and the information presented will be of great impact, for cell biologists and (electro)physiologists as well as clinicians. As mentioned by the authors, ²³Na MRI can be used to monitor the response to chemotherapy, and the expression level of voltage gated sodium channels (VGSCs) may serve as a biomarker, because VGSC expression correlates with malignancy/poor prognosis. The authors refer to work from their own, Mustafa Djamgoz's and Sébastien Roger's/Jean-Yves LeGuennec's group, i.e. those groups who have been fathering the concept of Na⁺ and VGSCs in breast cancer. They also describe VGSCs subunits as adhesion molecules and how VGSC activity can promote MMP-dependent invasiveness. In this context Leslie & Brackenbury do mention the interplay between VGSCs, NHE1 and the resulting extracellular acidification promoting cathepsin activity, however, the molecular mechanism remains vague by which increased Na⁺ influx through VGSCs (causing a decrease in the Na⁺ gradient) would stimulate NHE1 (taking advantage of this Na⁺ gradient) while decreasing the "driving force" for it. It would be nice if the authors could provide more details in addition to what they mention on top of page 8: "...via an allosteric interaction...". In the original paper by Lucie Brisson et al. (2013, J Cell

Sci) is written: "The change in cooperativity strongly suggests that Nav1.5 function in cancer cells allosterically increases NHE-1 activity in neutral ranges of pHi". - We have learnt more about the molecular structure of NHE1 and its molecular interactions since then. Is there any new knowledge of possible molecular switches? Brisson et al. (2013) conclude from a decrease in pHi-dependent Li⁺-uptake in shNav1.5 breast cancer cells that Nav1.5 promotes NHE1 activity. How do they exclude that Nav1.5 or other channels do not let pass Li⁺?

Response:

The Referee raises an important point regarding the mechanism by which Na⁺ influx through VGSCs could stimulate NHE1. This mechanism is still unclear. However, it is possible that Nav1.5 activity could increase NHE1 H⁺ extrusion via increased Na⁺/K⁺ activity and consequent glycolytic flux leading to a reduction in cytosolic pH. We have added this possibility to the manuscript on page 9.

In response to the comment regarding Li⁺, the Referee is correct in proposing that the findings of Brisson et al. could be explained by Li⁺ conductance through other proteins. Indeed, Bertil Hille showed that VGSCs are approximately as permeable to Li⁺ as to Na⁺ (Hille, 1972). Also, VGSCs are pH sensitive, so would be expected to increase Li⁺ uptake at lower pH. We have added this point into the manuscript on page 9.

- In addition to the graphical abstract the authors have prepared two nice figures but do not refer to them in the text body.

Response:

Thank you for pointing this out. We have now referred to these figures in the text.

- Page 3, graphical abstract: May I suggest to "reverse" the arrow pointing from "Intracellular [Na⁺]" to NHE1, so that it points from NHE1 to [Na⁺]_i? Actually, an increase in [Na⁺]_i (assuming constant pH) should slow down NHE1 activity.

Response:

This arrow is pointing from [Na⁺]_i to NHE1 because it symbolises the findings of Gillet et al 2009, and Brisson et al 2011 and 2013 arguing that Nav1.5 leads to increased NHE1 activity. The effect is blocked by TTX, suggesting that Na⁺ influx affects NHE1 activity. However, in response to the Referee's comment, we have added an additional arrow in the other direction since NHE1 activity will increase [Na⁺]_i. We have also added text in the figure legend to explain the two arrows.

- same as above: What would be the driving force for NHE1 if [Na⁺]_i in- and pHe decreased? I bring up this point once again because the authors state on the bottom of page 6: "Despite the need to maintain a Na⁺ gradient across the plasma membrane, several Na⁺ channels are upregulated in cancer, allowing further Na⁺ entry into cancer cells and >>depleting the Na⁺ gradient<<.

Response:

At the approximate resting membrane potential of the MDA-MB-231 cell line (-15 mV), and assuming a physiologically normal [Na⁺]_i of 5 mM, the Nernst equation would predict a driving force for Na⁺ of -104.8 mV. This would reduce to -67.7 mV at an elevated [Na⁺]_i of 20 mM (a conservative estimate based on available literature evidence). Thus, the inward Na⁺ driving force is reduced but not completely depleted when there is a small increase in [Na⁺]_i, and so Na⁺-dependent H⁺ export via NHE1 would still be expected to occur.

- Page 5, bottom: "of particular note are the Na⁺/H⁺ exchanger, NHE1, and the Na⁺/bicarbonate cotransporter, NBCn1, both of which are upregulated and highly active in breast cancer (Cardone et al.,.....). This information and the references could be extended by a more recent publication from the Boedtkjer-group (Acid-base transporters and pH dynamics in human breast carcinomas predict proliferative activity, metastasis, and survival. Toft NJ, Axelsen TV, Pedersen HL, Mele M, Burton M, Balling E, Johansen T, Thomassen M, Christiansen PM, Boedtkjer E. *Elife*. 2021 Jul 5;10:e68447. doi: 10.7554/eLife.68447). In this paper, Toft and colleagues draw a more detailed and differentiated picture of NHE1 and NBCn1 in the different types of breast cancer.

Response:

We agree that that is an important reference, so it has been added.

- Could a low salt diet represent a good adjuvant therapy?

Response:

This is an interesting question, but we don't believe that there is enough evidence to suggest this yet. Regulation of blood pressure and supporting kidney function are likely to be more important factors in cancer patients.

- page 8, 2nd paragraph, 1st sentence: should the comma be kept between "small" and "transmembrane"?

Response:

Thank you for pointing this out. It has been removed.

- Figure legend 1C, last line: "...causes an increaseD extracellular..." or "causes an increase IN extracellular...".

Response:

Thank you for pointing this out. The word "in" has been added.

Referee #2:

Leslie and Brackenbury present a concise review on sodium (channels) in breast cancer. This is a timely topic with a clear prospect for translation, and the text is well written so that there are only a few points that deserve further attention/clarification.

pg. 6, 1st para: The authors refer to the salt-inducible kinase SIK1. Is there any knowledge about this kinase in (breast) cancer with its elevated Na⁺ concentration? Is it pH sensitive?

Response:

Interestingly, SIK1 is normally downregulated in breast cancer. SIK1 activation stimulates oxidative phosphorylation and inhibits glycolysis. If the gene were not downregulated in cancer, then it would lead to increased oxidative phosphorylation when intracellular [Na⁺] is elevated (Ponnusamy & Manoharan, 2021). A sentence has been added to the paragraph on page 6 to address this point.

As for links between SIK1 and VGSCs, downregulation of SIK1 increases VGSC-dependent invasiveness in breast cancer (Gradek *et al.*, 2019). We have added discussion on this point to the beginning of the last paragraph on page 9.

pg. 7, ln4...: The authors cite several papers reporting upregulated expression of ENaC. Here a better distinction should be made between findings made in cell lines and those made in patient samples. Moreover, some of the papers only provide expression and no functional data.

Response:

This point has been clarified in the text on page 7.

pg. 8, 3rd para: The authors state that small GTPases are regulated by membrane potential depolarization. When looking up the papers I was surprised to see that the described depolarization is quite small, only a few millivolt. Can the authors speculate by which mechanism the small electrical signal is transduced into a biochemical one?

Response:

The membrane potential of MDA-MB-231 cells is very depolarised, at only -19 mV (Fraser *et al.*, 2005). Therefore a 4 mV change in membrane potential signifies a large proportional change in voltage. This would be expected to affect interactions between charged molecules at the membrane, specifically phosphatidylserine in the inner leaflet, leading to nanoclustering of anchored small GTPases and their subsequent activation (Zhou *et al.*, 2015). This point has been added to the manuscript.

pg. 9, top: It is not quite intuitive that an increase of Na⁺ should increase NHE1 activity. In terms of driving force, the opposite should be the case. Please discuss in more detail.

Response:

We agree that this is not intuitive and have added a sentence to that effect on page 9. As detailed in our response to Referee 1 above, we have also suggested a possible mechanism by which Na_v1.5 activity might increase NHE1 H⁺ extrusion via increased Na⁺/K⁺ activity and glycolytic flux.

References

- Fraser SP, Diss JKJ, Chioni AM, Mycielska ME, Pan HY, Yamaci RF, Pani F, Siwy Z, Krasowska M, Grzywna Z, Brackenbury WJ, Theodorou D, Koyuturk M, Kaya H, Battaloglu E, De Bella MT, Slade MJ, Tolhurst R, Palmieri C, Jiang J, Latchman DS, Coombes RC & Djamgoz MBA. (2005). Voltage-gated sodium channel expression and potentiation of human breast cancer metastasis. *Clin Cancer Res* **11**, 5381-5389.
- Gradek F, Lopez-Charcas O, Chadet S, Poisson L, Ouldamer L, Goupille C, Jourdan ML, Chevalier S, Moussata D, Besson P & Roger S. (2019). Sodium Channel Na_v 1.5 Controls Epithelial-to-Mesenchymal Transition and Invasiveness in Breast Cancer Cells Through its Regulation by the Salt-Inducible Kinase-1. *Sci Rep* **9**, 18652.
- Hille B. (1972). The permeability of the sodium channel to metal cations in myelinated nerve. *J Gen Physiol* **59**, 637-658.

Ponnusamy L & Manoharan R. (2021). Distinctive role of SIK1 and SIK3 isoforms in aerobic glycolysis and cell growth of breast cancer through the regulation of p53 and mTOR signaling pathways. *Biochim Biophys Acta Mol Cell Res* **1868**, 118975.

Zhou Y, Wong CO, Cho KJ, van der Hoeven D, Liang H, Thakur DP, Luo J, Babic M, Zinsmaier KE, Zhu MX, Hu H, Venkatachalam K & Hancock JF. (2015). Membrane potential modulates plasma membrane phospholipid dynamics and K-Ras signaling. *Science* **349**, 873-876.

Dear Dr Brackenbury,

Re: JP-SR-2022-282306R1 "Sodium channels and the ionic microenvironment of breast tumours" by Theresa K Leslie
William J Brackenbury

I am pleased to tell you that your Symposium Review article has been accepted for publication in The Journal of Physiology, subject to any modifications to the text that may be required by the Journal Office to conform to House rules.

NEW POLICY: In order to improve the transparency of its peer review process The Journal of Physiology publishes online as supporting information the peer review history of all articles accepted for publication. Readers will have access to decision letters, including all Editors' comments and referee reports, for each version of the manuscript and any author responses to peer review comments. Referees can decide whether or not they wish to be named on the peer review history document.

The last Word version of the paper submitted will be used by the Production Editors to prepare your proof. When this is ready you will receive an email containing a link to Wiley's Online Proofing System. The proof should be checked and corrected as quickly as possible.

All queries at proof stage should be sent to tjp@wiley.com

The accepted version of the manuscript is the version that will be published online until the copy edited and typeset version is available. Authors should note that it is too late at this point to offer corrections prior to proofing. Major corrections at proof stage, such as changes to figures, will be referred to the Reviewing Editor for approval before they can be incorporated. Only minor changes, such as to style and consistency, should be made a proof stage. Changes that need to be made after proof stage will usually require a formal correction notice.

Are you on Twitter? Once your paper is online, why not share your achievement with your followers. Please tag The Journal (@jphysiol) in any tweets and we will share your accepted paper with our 22,000+ followers!

Yours sincerely,

Dr Peiyong Fong
Senior Editor
The Journal of Physiology
<https://jp.msubmit.net>
<http://jp.physoc.org>
The Physiological Society
Hodgkin Huxley House
30 Farringdon Lane
London, EC1R 3AW
UK
<http://www.physoc.org>
<http://journals.physoc.org>

REQUIRED ITEMS:

-It is the authors' responsibility to obtain any necessary permissions to reproduce previously published material
https://jp.msubmit.net/cgi-bin/main.plex?form_type=display_requirements#use

Comments:

Reviewing Editor:

No further comments

Senior Editor:

Thank you for thoroughly responding to all points raised in initial review of your manuscript. Both Expert Referees agree this is a balanced and exceptionally well-written treatment of the highly important topic of sodium homeostatic pathways in

tumors. The Reviewing Editor and I concur with their assessment. For those--who attended virtually the senior author's talk at last year's meeting, this serves as an excellent summary of the presentation. In addition, I anticipate it will serve as a roadmap for future work in the field, and therefore be highly impactful. Well done.

REFeree COMMENTS:

Referee #1:

In their revised manuscript the authors have addressed all the points raised in a proper and most satisfying way. I have no further comments.

Referee #2:

All questions have been answered satisfactorily. I have no further concerns.

* IMPORTANT NOTICE ABOUT OPEN ACCESS *

To assist authors whose funding agencies mandate public access to published research findings sooner than 12 months after publication The Journal of Physiology allows authors to pay an open access (OA) fee to have their papers made freely available immediately on publication.

You will receive an email from Wiley with details on how to register or log-in to Wiley Authors Services where you will be able to place an OnlineOpen order.

You can check if you funder or institution has a Wiley Open Access Account here <https://authorservices.wiley.com/author-resources/Journal-Authors/licensing-and-open-access/open-access/author-compliance-tool.html>

Your article will be made Open Access upon publication, or as soon as payment is received.

If you wish to put your paper on an OA website such as PMC or UKPMC or your institutional repository within 12 months of publication you must pay the open access fee, which covers the cost of publication.

OnlineOpen articles are deposited in PubMed Central (PMC) and PMC mirror sites. Authors of OnlineOpen articles are permitted to post the final, published PDF of their article on a website, institutional repository, or other free public server, immediately on publication.

Note to NIH-funded authors: The Journal of Physiology is published on PMC 12 months after publication, NIH-funded authors DO NOT NEED to pay to publish and DO NOT NEED to post their accepted papers on PMC.